# Mixing of meteoric and geothermal fluids supports hyperdiverse chemosynthetic hydrothermal communities

Daniel R. Colman[1], Melody R. Lindsay[1] & Eric S. Boyd [1,2]

Little is known of how mixing of meteoric and geothermal fluids supports biodiversity in non-photosynthetic ecosystems. Here, we use metagenomic sequencing to investigate a chemosynthetic microbial community in a hot spring (SJ3) of Yellowstone National Park that exhibits geochemistry consistent with mixing of a reduced volcanic gas-influenced end member with an oxidized near-surface meteoric end member. SJ3 hosts an exceptionally diverse community with representatives from ~50% of known higher-order archaeal and bacterial lineages, including several divergent deep-branching lineages. A comparison of functional potential with other available chemosynthetic community metagenomes reveals similarly high diversity and functional potentials (i.e., incorporation of electron donors supplied by volcanic gases) in springs sourced by mixed fluids. Further, numerous closely related SJ3 populations harbor differentiated metabolisms that may function to minimize niche overlap, further increasing endemic diversity. We suggest that dynamic mixing of waters generated by subsurface and near-surface geological processes may play a key role in the generation and maintenance of chemosynthetic biodiversity in hydrothermal and other similar environments.

[1] Department of Microbiology and Immunology, Montana State University, Bozeman, MT 59718, USA. [2] NASA Astrobiology Institute, Ames Research Center, Mountain View, CA, USA. Correspondence and requests for materials should be addressed to E.S.B. (email: eboyd@montana.edu)

Mixing of subsurface- and surface-derived fluids defines Earth's habitable (or critical) zone, which represents the outermost layers of the continental crust that are affected by atmospheric, geological, and hydrological processes. The most studied portions of the critical zone are in the near-surface (vadose zone) that are heavily influenced by photosynthetic productivity. However, the critical zone extends empirically to depths up to ~5 km[1] and could extend to depths exceeding 12 km[2] where influences from photosynthesis are likely to be minimal and where life is likely constrained by a combination of high temperature and nutrient or energy limitation[3,4]. Numerous studies have documented the presence of microbial communities in subsurface environments[1,5–7]; however, little is known of how the confluence of subsurface and surface-associated processes promotes the generation and maintenance of biodiversity in subsurface, chemosynthetic systems. This is due largely to difficulties in accessing these ecosystems and isolating fluid types that source these environments.

Continental hydrothermal systems integrate waters from the deep subsurface and surface to generate extraordinarily diverse geochemical compositions in accessible terrains[8]. At temperatures > ~70 °C in circumneutral to alkaline (pH > 6.5) springs or > 54 °C in acidic (pH < 3.5) springs, photosynthesis is excluded and microbial life is supported by chemosynthesis[9]. At first order, extremes in temperature constrain microbial diversity in chemosynthetic hot springs[10,11]. However, studies conducted in thermal springs have revealed chemosynthetic communities that are exceptionally diverse[12–14], while others have revealed communities that comprise as few as one or two populations, despite similar spring temperatures[14]. These discrepancies suggest geochemical controls on the generation and maintenance of chemosynthetic hot spring community biodiversity.

Thermal springs have served as integral platforms for examining the role of geochemical variation in shaping microbial diversity and have provided new insights into novel forms of microbial life and the processes that sustain this life[14–16]. Indeed, some of the earliest environmental studies of uncultivated microbial diversity were conducted in thermal springs like Obsidian Pool, Yellowstone National Park (YNP), Wyoming, leading to the recognition of dozens of new archaeal and bacterial taxonomic divisions from this single spring[12,13]. Likewise, recent environmental genomic surveys from YNP hot spring communities have yielded new insights into the genomic and physiological diversity of uncultivated lineages including those thought to represent a missing link to proto-Eukaryotes[17], those that serve as analogs to cosmopolitan subsurface taxa[18], and have provided insights into some of the smallest known microbial symbionts[19]. Many of these newly described lineages contribute to our understanding of the physiological limits of life on Earth[10,20] and have served as analogs for understanding the capacity for life in the deep subsurface of Earth[21]. Yet, little investigation has been conducted to discern how the generation and maintenance of such diversity is related to the geological processes that influence geochemical variation in hydrothermal systems. In other words, considerable research has attempted to identify what types of diversity are present in hydrothermal systems, but little attention has been directed towards understanding how the diversity arises and how it is differentially promoted across spring types.

The geochemistry of hot spring waters is controlled by interaction between subsurface and surface geologic processes[22]. Paramount among these is separation of waters into a gas-poor, circumneutral liquid phase that is enriched in non-volatile solutes (e.g., $Cl^-$) and a gas-rich vapor phase that is enriched in reduced gases such as $H_2S$, $CH_4$, and $H_2$[22–24]. Gas-rich vapor can condense and mix with $O_2$-rich meteoric waters as it ascends to the surface, promoting the oxidation of $H_2S$ and the subsequent generation of acidic springs[23,25]. Broadly these two geothermal fluids result in two geothermal end member water types with circumneutral (pH ~6.5–7.5) and acidic (pH ~2.5–3.5) pH[23,25]. These two geothermal water types, along with precipitation-derived or meteoric waters, represent three end member water compositions that broadly define hot spring geochemistry[23]. The microbial communities associated with these end member compositions have been characterized extensively[26]. However, dynamic mixing of these end members can generate spring waters that are intermediate in composition between the end member waters described above (i.e., pH ~4.0 ~ 6.0). Far less is known about the microbial communities inhabiting springs sourced by mixed end member waters.

Here, we describe a hyperdiverse chemosynthetic thermal spring community from YNP termed 'Smokejumper 3' (SJ3; pH = 5.4, $T$ = 61.9 °C), which hosts an extensive diversity of archaeal and bacterial lineages. The geochemical composition of SJ3 waters represent an extreme example of mixing of a reduced vapor phase-influenced water enriched in volcanic gases with oxidized near-surface meteoric water. Many of the lineages in SJ3 have not previously been characterized via genomic analyses, nor have several identified functionalities (e.g., methanogenesis) been previously identified in the lineages that are present within the SJ3 community. Comparison of the SJ3 metagenome with other available YNP chemosynthetic metagenomes indicates that spring waters with geochemical compositions underpinned by similar geological processes also host exceptionally diverse microbial communities that are functionally similar to that of SJ3. The results are discussed in the context of how dynamic spatial and temporal mixing of water compositions likely support and maintain biodiverse chemosynthetic communities in hydrothermal settings.

## Results

**Geologic and geochemical context of Smokejumper geyser basin**. The Smokejumper geyser basin (SJGB) is located in Southwestern Yellowstone National Park (YNP), Wyoming, at a topographic high in the Summit Lake Rhyolite (lava) Flow (Fig. 1). The SJGB is also located at high elevation, and near the continental hydrological divide of the Americas and on the edge of the Yellowstone caldera boundary (Fig. 1), where extensive-faulting and fracturing of volcanic bedrock is thought to promote the release of gases that were exsolved during subsurface phase separation[27]. SJ3 is a gas-rich, hydrothermal spring located within the SJGB (N 44°24'57.42"; W -110°57'20.76"). SJ3 waters exhibit low conductivity (0.42 mS, Supplementary Fig. 1) and a pH of 5.4 (Supplementary Fig. 2), characteristics that have been suggested to result from mixing of vapor phase gas-influenced waters with meteoric waters[26,28]. Moreover, the very low chloride concentrations measured in SJGB springs (~1.0–1.1 mg $L^{-1}$)[24] suggest minimal to no input of waters from the deep hydrothermal reservoir[22,23], while $SO_4^{2-}$ concentrations are elevated (~54–70 mg $L^{-1}$; Supplementary Fig. 3) relative to meteoric waters[24], likely due to the oxidation of vapor phase-sourced sulfide (Supplementary Fig. 3). Taken together, the low conductivity, slightly acidic pH, low $Cl^-$, and moderate levels of $SO_4^{2-}$ suggest low residence time of meteoric waters. Phase separated volcanic gas-influenced geothermal fluids are often enriched in $H_2$, CO, and short-chain alkanes due to interaction between geothermal fluids and ferrous iron containing minerals in bedrock during ascension of the gases and waters to the surface[24]. Consistent with this observation, dissolved $H_2$ and $CH_4$ concentrations in springs of the Smokejumper geyser basin are among the highest concentrations reported in hot spring waters in YNP[24].

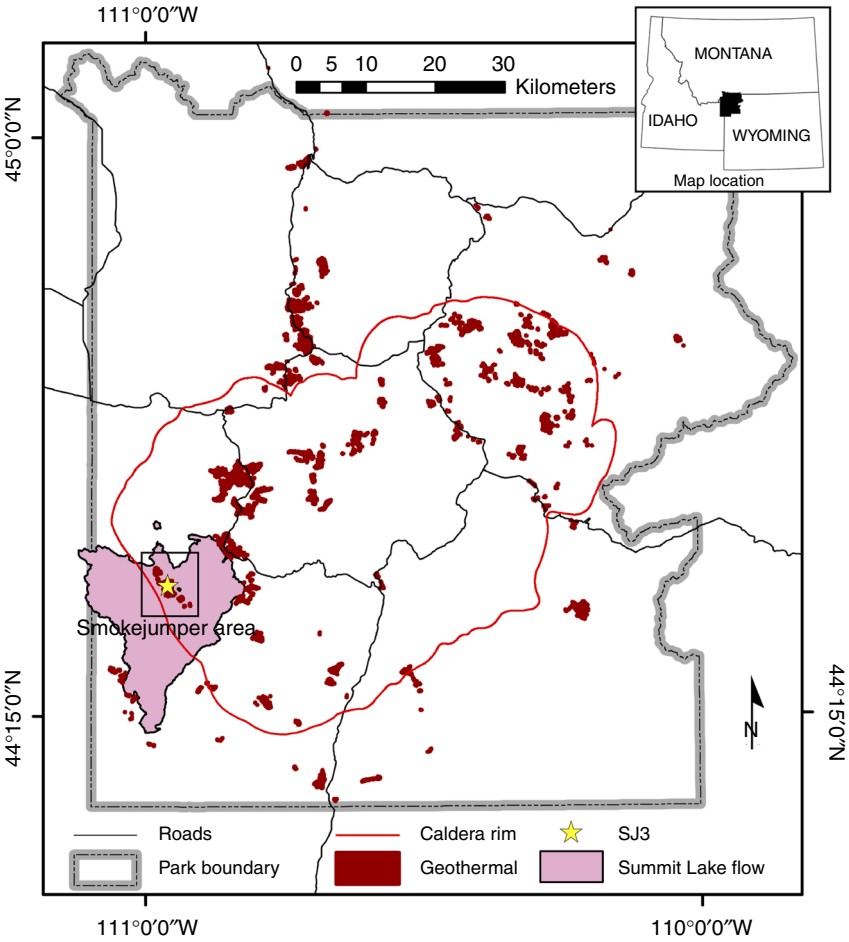

**Fig. 1** Map of Yellowstone National Park (YNP) showing the location of the Smokejumper Geyser Basin (SJGB) and 'Smokejumper 3' spring (SJ3). The caldera rim, road reference layers, park boundary, and Summit Lake Rhyolite Flow reference is the same as in Christiansen (2001)[27]

**High taxonomic and functional diversity of the SJ3 community**. Assembly of contigs from the SJ3 metagenome (~61 Gbp of reads, 278 Mbp assembled length, Supplementary Table 1) and subsequent binning of genomes into metagenome-assembled genomes (MAGs) based on tetranucleotide frequencies and coverage profiles resulted in 108 draft MAGs. Of these 108 MAGs, 82 were estimated to be > 50% complete and exhibited < 7% contamination, meeting currently accepted criteria as medium to high quality draft MAGs[29] (Fig. 2, Supplementary Data 9). Populations closely related to the chemolithoautotrophic $H_2S$/$S_2O_3^-$/$S^0$ oxidizing *Sulfurihydrogenibium yellowstonense* (phylum: Aquificae) and chemoheterotrophic $S^0$/$S_2O_3^-$/$SO_3^{2-}$ reducing *Caldisericum exile*[30] (phylum: Caldiserica; formerly candidate division OP5) dominate the community (~15% relative abundance each; Supplementary Data 9), which is consistent with the prevalence of *Sulfurihydrogenibium* populations in weakly acidic YNP hot springs[31]. The remainder of the genomes comprised diverse low abundance populations with 10 and 23 archaeal order/phylum-level and bacterial phylum-level groups present, respectively, and with multiple representatives within many of the groups (Fig. 3, Supplementary Data 9).

Populations within the SJ3 community represented ~50% of the known higher-order lineages within both Archaea (order/class level) and Bacteria (phylum level) (Fig. 3). This level of diversity is particularly high for chemosynthetic hydrothermal system sediment/water communities that have previously been shown to typically comprise only a few lineages[14,32,33]. Further,

SJ3 genomes contributed 8% of the branch length to both archaeal and bacterial phylogenetic trees inclusive of all the major lineages for each domain. While the total branch length and taxonomic breadth within a phylogeny is dependent on i) the taxa that are included in the analysis, ii) lineage representation in available databases, and iii) lineage definitions[34,35], these analyses nevertheless suggest that this hot spring community is hyperdiverse at the taxonomic and phylogenetic levels when compared to previously analyzed hot spring communities.

A number of the lineages observed in SJ3 are underrepresented in other hot springs[14,31–33,36,37], including many that have only rarely been observed in YNP hot springs such as the *Candidatus* (*Ca.*) 'Acetothermia', Archaeoglobales, *Ca.* 'Bathyarchaeota', Deep sea Hydrothermal Vent Euryarchaeota group 2 (DHVE2), DPANN-related lineages, *Ca.* 'Hadesarchaea', Thaumarchaeota, and *Ca.* 'Verstraetarchaeota', among others. Phylogenomic reconstructions indicate that several SJ3 MAGs represent the currently deepest-branching genomic representatives of *Ca.* 'Aigarchaeota', Korarchaeota, Thaumarchaeota, *Ca.* 'Bathyarchaeota', *Ca.* 'Verstraetarchaeota', and the DHVE2 (Supplementary Data 1), suggesting that they may provide new insight into the divergence of these archaeal groups. Indeed, several of these deep-branching lineages belong to monophyletic clades that are phylogenetically distant from other characterized members of their respective lineages (e.g., as in the Thaumarchaeota, Korarchaeota, and *Ca.* 'Aigarchaeota'), and thus may comprise lineages distinct from those indicated in Fig. 3.

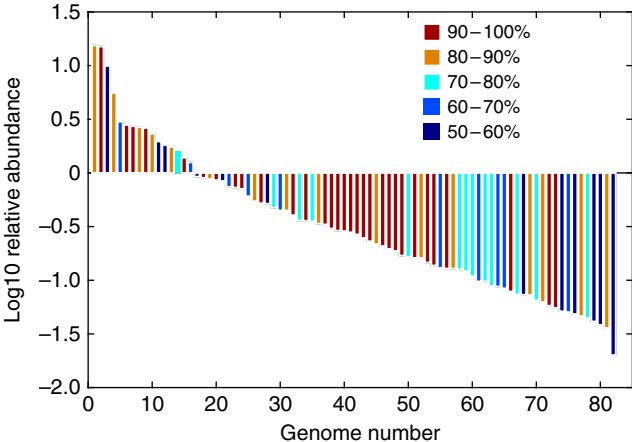

**Fig. 2** Rank-abundance plot of SJ3 reconstructed population level bins. Each vertical bar represents a reconstructed genome bin that has an estimated completeness > 50% ($n = 82$). Genome bins are arranged by $\log_{10}$ transformed relative abundance (as a percentage) in decreasing order, as determined by read mapping. The scale on the upper right corresponds to the estimated percent completeness of each genome bin

**End member fluid mixing and chemosynthetic diversity.** To investigate whether the functional coding potential within the SJ3 metagenome was high among chemosynthetic YNP hot spring communities in general, protein family diversity was compared among SJ3 and publicly available YNP chemosynthetic spring community metagenomes. Rarefaction analysis and random subsampling of protein families in chemosynthetic communities from SJ3 and other publicly available YNP hot spring communities ($n = 48$; Supplementary Data 10) revealed that the SJ3 community comprises one of the highest levels of protein diversity among metagenomically characterized chemosynthetic YNP spring communities (Fig. 4). This result held with increasing subsampling depth (Fig. 4) indicating that the deep sequencing of SJ3 (298,127 proteins in the analysis) did not inflate protein family diversity relative to less-thoroughly characterized YNP metagenomes (e.g., those produced with Sanger sequencing). Indeed, several metagenomes generated from less total sequence when compared to SJ3 yielded similarly high levels of protein diversity even when compared at the level of only 10,000 subsampled proteins (Fig. 5a; Supplementary Data 10). Two of these metagenomes, 'Obsidian Pool Prime', and 'Washburn Spring', were produced with Sanger sequencing[14] and comprised >2 orders of magnitude less assembled sequence data than SJ3, suggesting that sequencing effort, per se, did not preclude comparisons of overall protein diversity when selecting random subsets of rarefied proteins from each metagenome.

A significant linear relationship (adjusted $R^2 = 0.33$, $p < 0.001$, $n = 41$; Fig. 5) was observed when comparing protein-coding gene diversity (after subsampling at 10,000 random proteins) against the ratio of $SO_4^{2-}/Cl^-$ of waters in springs where these data were available (Supplementary Data 10). Spring waters with elevated $SO_4^{2-}/Cl^-$ ratios, such as SJ3, are interpreted to reflect mixing of vapor phase (contributing elevated $SO_4^{2-}$ from $H_2S$ oxidation) with near-surface meteoric waters that lack significant $Cl^-$ input from the deeply-sourced hydrothermal reservoir[23]. Conversely, spring waters with low $SO_4^{2-}/Cl^-$ ratios are indicative of less input of vapor phase and are interpreted to reflect input from the deep hydrothermal reservoir[23,38]. The large amount of variation unaccounted for in the linear model can be attributed to the role of other variables and processes that are unaccounted for in this model that may influence the functional diversity of spring communities. This includes temperature and

the temporal dynamics of the spring, both of which have been suggested to influence the biodiversity of thermal springs[11,33,39,40]. Regardless, the prevalence of higher protein-coding gene diversity in communities inhabiting springs that are sourced by mixing of end member fluids indicates that this process promotes functional diversity in thermal springs, and likely also contributes to the elevated taxonomic diversity observed in these springs[12,13]. Consistent with this interpretation, several of the communities that encoded higher protein-coding gene diversity than SJ3, and which had published geochemical information on spring waters (i.e., 'Washburn Spring', 'Obsidian Pool Prime', and Obsidian Pool (*sensu stricto*)), all shared similar attributes with SJ3 including hosting waters with similar pH, high $SO_4^{2-}$, and low $Cl^-$ concentrations (Fig. 5, Supplementary Data 10). This observation suggests that the geologic processes that define the geochemical composition of these spring types (i.e., end member water mixing) is potentially also involved in supporting and maintaining the high levels of taxonomic and functional biodiversity in these springs.

While the role of geothermal fluid mixing in supporting community diversity has not been extensively investigated in continental hydrothermal systems, several investigations of marine hydrothermal vent systems suggest that fluid mixing influences the diversity of vent communities[41–44]. Indeed, mixing of high temperature hydrothermal fluids (>200 °C) with cold seawater promotes chemical disequilibrium in redox reactions that can support microbial metabolism[45], likely leading to increased niche space capable of supporting higher diversity. Although there is considerable evidence that mixing of hydrothermal fluids with seawater influences community diversity, to the best of our knowledge, an explicit comparison of marine hydrothermal vent community diversity levels with chemical proxies that can be used to deduce the extent of fluid mixing has not been conducted. Nevertheless, recent metagenomic analyses have revealed high levels of metabolic and taxonomic diversity in marine sediments within environments hosting fluids that are reflective of considerable hydrothermal fluid/seawater mixing at the Juan de Fuca Ridge Flank[46], as well as in the Guaymas Basin[47]. Consequently, it is likely that mixing of end member fluids leads to similar ecological outcomes in both marine and continental hydrothermal systems.

**Subsurface-sourced gases support SJ3 community metabolism.** To compare SJ3 functional potential to other chemosynthetic YNP metagenomes and investigate the links between geothermal fluid mixing and SJ3 metabolic diversity, the metabolic protein-coding differences between SJ3 and a subset ($n = 14$) of available YNP metagenomes were compared. The encoded protein complements involved in energy metabolism in SJ3 populations were like those from 'Obsidian Pool Prime' (OPP_17) and 'Washburn Spring' (WS_18). Together, these communities formed a distinct cluster in a PCoA ordination of inferred protein composition dissimilarities (Fig. 6a). The encoded proteins that separated these three springs (particularly SJ3) from other springs analyzed included [NiFe]-hydrogenases ($F_{420}$ hydrogenase, Frh; NADP-coupled hydrogenase, Hyh; bidirectional hydrogenase, Hox), carbon monoxide dehydrogenase (Coo), carbonic anhydrase (Cah), formate dehydrogenase (Fdh), and those associated with methanogenesis/anaerobic alkane oxidation (methyl coenzyme-M reductases, Mcr; tetrahydromethanopterin S-methyltransferase, Mtr; formylmethanofuran dehydrogenase, Fwd) (Fig. 6b, Supplementary Data 3). In addition, genes encoding homologs of acetate kinases (Ack) and acetate synthases (Acs) putatively involved in acetogenesis[48] were also among the more enriched protein-coding genes in SJ3.

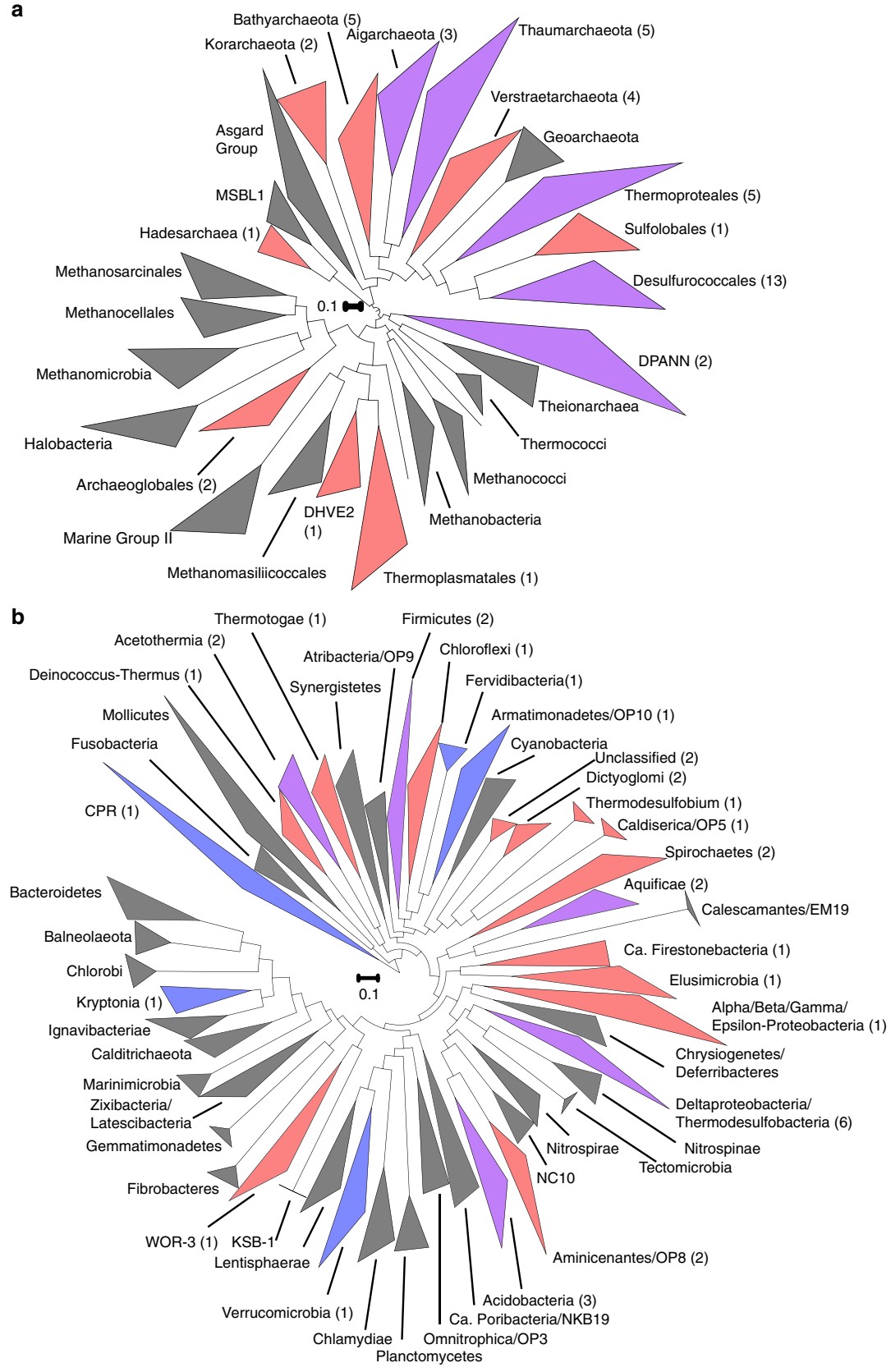

**Fig. 3** Maximum likelihood phylogenetic reconstruction of archaeal and bacterial population level MAGs from SJ3. Trees representing the major **a** archaeal order- (or class/phylum-) level clades and the **b** bacterial phylum-level clades were constructed using up to 104 or 31 phylogenetic marker genes, respectively (>50% in any genome). Clades are collapsed as triangles with the taxonomic designation provided to the right. The numbers of MAGs that were identified in a specified clade are indicated in parentheses. Clades encompassing MAGs from 'SJ3' that were >50% complete are shown in blue, those with MAGs >75% are shown in red, and those representing multiple MAGs that ranged in completeness from 50 to >75% are shown in purple. The scale bar shows the expected number of substitutions per site. The phylogenetic trees without collapsed clades showing individual MAG placement are provided as Supplementary Data 1 (Archaea) and Supplementary Data 2 (Bacteria)

Genes encoding putative Mcr proteins were identified in SJ3 MAGs associated with the *Ca.* 'Verstraetarchaeota' (SJ3.Bin27 and SJ3.Bin82.1.1) and Archaeoglobales (SJ3.Bin34). Mcr protein homologs have been recently identified in MAGs of putatively methanogenic (or alkanotrophic) lineages outside of the canonical euryarchaeotal methanogen groups, including representatives of the *Ca.* 'Bathyarchaeota' and *Ca.* 'Verstraetarchaeota'[49,50]. Unlike previous analyses of *Ca.* 'Bathyarchaeota' from coal bed methane wells[49], no evidence for methanogenesis or methanotrophy was identified in the five phylogenetically distinct (Supplementary Data 1) *Ca.* 'Bathyarchaeota' MAGs that were present in the SJ3 community. Regardless, the enrichment of genes coding for proteins putatively involved in the metabolism of substrates that are abundant in volcanic gases (e.g., $H_2$, $CH_4$ or other alkanes, CO), and that are particularly abundant in SJ3[24], is consistent with the SJ3 community being adapted to take advantage of these substrates as electron donors and/or carbon sources. Many of the genes coding for the above-mentioned functionalities were also enriched in the 'Obsidian Pool Prime' and 'Washburn Spring' metagenomes (Fig. 6b, Supplementary Data 3), which are likely to be sourced by vapor phase-influenced waters (and/or subsurface-sourced gases) mixing with meteoric waters, based on similar geochemical profiles as SJ3[14,51] (Fig. 5, Supplementary Data 10).

**Metabolic differentiation also promotes high diversity**. The detection of closely related genome bins within the diverse SJ3 community prompted investigation of putative ecological mechanisms that may contribute to the maintenance of such high levels of diversity in mixed fluid systems and whether such diversity could be related to the process of end member fluid mixing. Specifically, we sought to determine whether metabolic differentiation of taxa (i.e., the minimization of niche overlap) belonging to the same higher-order taxonomic groups allows for population coexistence. Comparison of archaeal and bacterial populations belonging to the same lineages (and exhibiting > 75% completeness) were conducted for ten archaeal taxonomic groups (*Ca.* 'Aigarchaeota', Archaeoglobales, *Ca.* 'Bathyarchaeota', Desulfurococcales, Korarchaeota, Thaumarchaeota, Thermoplasmata, Thermoproteales, and *Ca.* 'Verstraetarchaeota'), and seven bacterial groups (Acidobacteria, *Ca.* 'Aminicenantes', Deltaproteobacteria, Dictyoglomi, Spirochaetes, Thermodesulfobacteria, and Dictyoglomi-like unclassified Bacteria) (Supplementary Data 9). Comparisons of G+C% content, coverage profiles, and phylogenetic distances among MAGs indicated that they represent distinct populations and were not artefacts of the genome binning approaches used here (Supplementary Data 4, Supplementary Data 5).

We first hypothesized that closely related MAGs that are differentiated due to the loss or acquisition of traits should exhibit a greater level of difference in the composition of metabolism-related protein-coding genes relative to genetic processing protein-coding genes. To test this hypothesis, we first compared differentiation in encoded proteins among genomes that were annotated within the KEGG category of 'metabolism' (which

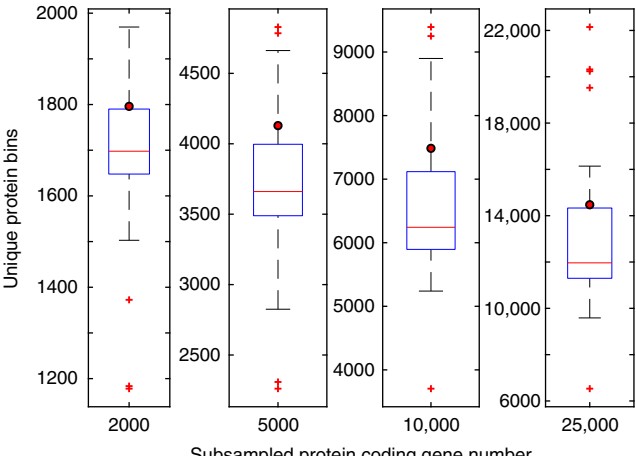

**Fig. 4** Comparison of protein family diversity in SJ3 relative to other chemosynthetic YNP hot spring metagenomes. Boxplots show the distribution of unique protein bins after subsampling 48 chemosynthetic YNP metagenomes at multiple depths at the levels shown below each box plot. Note that the Y-axis scales differ between panels. The values for SJ3 are shown as red circles. Red lines indicate the median for each distribution, while the boxes represent the interquartile range between the 25th and 75th percentiles. Whiskers show the full range of the data, not considering outliers and outliers are shown as red crosses. Source data are provided within the source data file accompanying this manuscript

primarily comprises proteins involved in energy conservation) vs. differentiation of encoded proteins among genomes involved in 'genetic information processing'. A greater extent of differentiation of metabolism-related proteins relative to genetic processing proteins was observed in all but one (Dictyoglomi SJ3.Bin31.1.2 vs. SJ3.Bin76.1.7) of the 49 pairwise comparisons, suggesting that metabolic differentiation was ubiquitous among SJ3 populations that belonged to the same taxonomic lineage (Fig. 7a).

Further comparison of the functional potentials among MAGs within the same taxonomic group indicates that closely related taxa are differentiated at the metabolic level, potentially allowing for their co-existence and the maintenance of high levels of diversity in SJ3 (Fig. 7). For example, a single Archaeoglobales-related MAG (SJ3.Bin34; 98.7% estimated completeness) with the putative capacity for methanogenesis, methanotrophy, or higher chain alkane oxidation (Supplementary Fig. 4, Supplementary Data 6), co-occurred with other Archaeoglobales (SJ3.Bin61; 89.5% estimated completeness) that exhibited metabolic potential that was more typical of other Archaeoglobales (i.e., sulfate/ sulfite/thiosulfate reduction[52]; Fig. 7c; Supplementary Data 6). The SJ3.Bin34 and SJ3.Bin61 MAGs are closely related phylogenetically, and encode proteomes that are 72% ± 13% identical at the amino acid level, suggesting that they likely belong to the same family- or genus-level clade[53]. SJ3.Bin34 forms an outgroup to SJ3.Bin61 and another unpublished MAG from a Chinese hot spring (*Archaeoglobus* sp. JZ bin_24; IMG genome

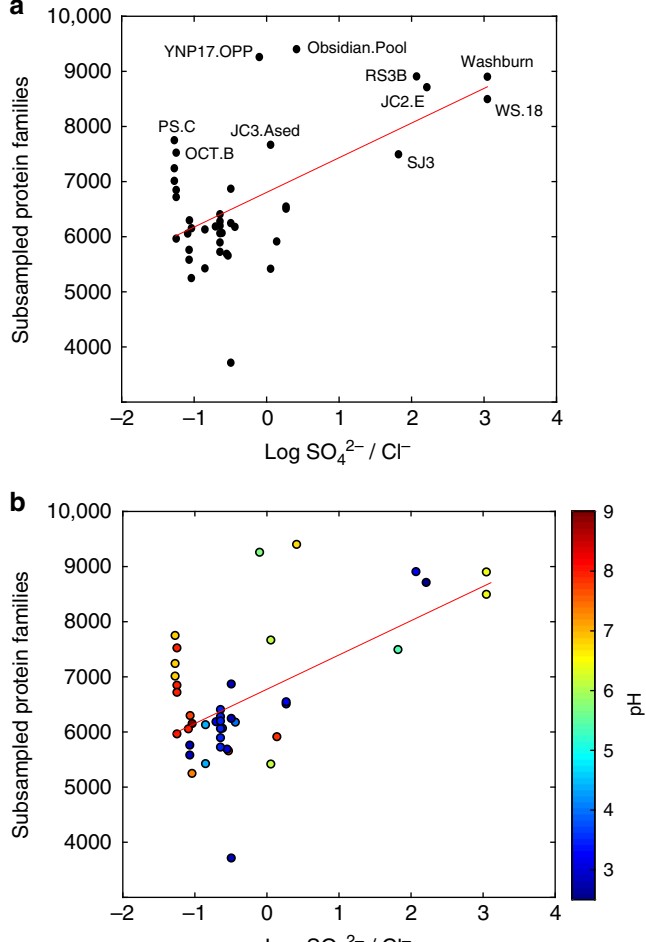

**Fig. 5** Comparison of subsampled protein family diversity in chemosynthetic YNP hot spring metagenomes and inferred spring water origins. Protein subsampling was conducted on 10,000 randomly chosen protein homologs from each metagenome. The log transformed $SO_4^{2-}/Cl^-$ ratio was calculated from publicly available data for each spring (Supplementary Data 10). Those metagenomes with higher protein family diversity than SJ3 after subsampling are indicated in **a**, with IDs corresponding to those provided in Supplementary Data 10. Red line shows linear regression model (adjusted $R^2 = 0.33$, $P < 0.001$, $n = 41$). The pH of the springs from which the specified metagenomes were sampled from is overlaid on a plot of subsampled protein families as a function of log transformed $SO_4^{2-}/Cl^-$ ratios in **b**. Source data are provided within the source data file accompanying this manuscript

ID: 2721755890). However, genes coding for proteins involved in methanogenesis or anaerobic alkane oxidation (e.g., McrAGCDB), in addition to numerous other methanogenesis-related accessory proteins, are only present in the SJ3.Bin34 MAG (Fig. 7c; Supplementary Figs. 4 and 5, Supplementary Data 6). Conversely, genes coding for key proteins involved in sulfate/sulfite/thiosulfate reduction (AprAB, DsrAB) are present in the SJ3.Bin61 MAG, but not that of SJ3.Bin34.

Intriguingly, the SJ3.Bin34 McrA homolog forms a basal lineage to Ca. 'Verstraetarchaeota' McrA homologs identified here and elsewhere[50] (Supplementary Fig. 6). Given that the SJ3.Bin34/SJ3.Bin61/JZ bin_24 clade is nested within other well-characterized sulfate/sulfite/thiosulfate-reducing Archaeoglobales (Fig. 7c), the genetic capacity for putative methane or alkane metabolism was either horizontally acquired in the SJ3.Bin34 lineage from a yet to be identified source, or otherwise retained as

an ancestral trait that was lost in other Archaeoglobales. Methylotrophic methanogenesis has been hypothesized in the Ca. 'Verstraetarchaeota'[50], although the lack of cultivation information precludes definitive assignment of this functionality. However, the phylogenetic association of the Archaeoglobales/Ca. 'Verstraetarchaeota' McrA homologs with McrA homologs from hydrogenotrophic methanogens such as Methanopyrus, Methanobacteriales, and Methanococcales[54] (Supplementary Fig. 6), leads us to cautiously speculate that the SJ3 Archaeoglobales population represented by the SJ3.Bin34 MAG may be involved in hydrogenotrophic methanogenesis. This assertion is consistent with the lack of protein homologs necessary for methylotrophic methanogenesis (Supplementary Data 6), and the presence of protein homologs necessary for hydrogenotrophic methanogenesis (Supplementary Fig. 4). Nevertheless, here and in other comparisons, it is possible that the lack of genome completeness could lead to the apparent absence of some functional gene homologs. However, this is unlikely for the SJ3.Bin34 MAG owing to its high estimated completeness (~99%) and other lines of evidence suggesting the potential for hydrogenotrophic methanogenesis. The capacity to produce small amounts of $CH_4$ in vitro has been previously documented for Archaeoglobus fulgidus[55], despite that Mcr protein-coding genes have not been previously observed in the genome of A. fulgidus or other published Archaeoglobales isolates or genomes. Thus, while other Archaeoglobales may be able to generate minor amounts of $CH_4$ in vitro, it must necessarily be via a mechanism unlike that of the energy conservation pathway in methanogens and potentially also the population represented by SJ3.Bin34. Regardless, this apparent transfer or retention of ancestral Mcr homologs in the Archaeoglobales lineage has likely allowed it to inhabit an available niche whose dimensions are defined by variable input of volcanically sourced gases such as $H_2$, $CO_2$, $CH_4$ or short-chain alkanes.

Potential metabolic differentiation was also observed in Ca. 'Verstraetarchaeota' MAGs ($n = 4$) wherein genes encoding McrABG were identified in later diverging Ca. 'Verstraetarchaeota' MAGs (SJ3.Bin27 and SJ3.Bin82.1.1; Supplementary Fig. 7) but were absent in earlier-diverging lineages represented by the MAGs SJ3.Bin21 and SJ3.Bin40 (Supplementary Data 7). The only published evidence of functional potential in the Ca. 'Verstraetarchaeota' derives from MAGs recovered from anaerobic digesters[50], wherein these populations were hypothesized to be capable of methylotrophic methanogenesis due to the presence of protein-coding genes necessary to utilize methanol (MtaA; [methyl-Co(III) methanol-specific corrinoid protein]:coenzyme M methyltransferase), methylamines (MtbA; [methyl-Co(III) methylamine-specific corrinoid protein]:coenzyme M methyltransferase, MtmB; methylamine-corrinoid protein Co-methyltransferase, MtbC; dimethylamine corrinoid protein), or methanethiols (MtsA; methylated-thiol-coenzyme M methyltransferase). Methylotrophy-associated homologs were present in SJ3.Bin27 (MtaA, MttC; trimethylamine corrinoid protein, MtmB) and SJ3.Bin82.1.1 (MtaA, MttC, MtmB, MtbC) (Supplementary Data 7), suggesting a similar metabolic potential as those described previously. In contrast, none of the above homologs were identified in SJ3.Bin21 or SJ3.Bin40, or other closely related, earlier-diverging Ca. 'Verstraetarchaeota' (Fig. 7e). This suggests relatively recent metabolic differentiation of the SJ3 Ca. 'Verstraetarchaeota' populations, potentially allowing them to capitalize on available $CH_4$, short-chain alkanes, or methylated compounds in SJ3 (Fig. 7e). Moreover, the capacity for carbon monoxide (CooS) or formate (FdhF) utilization was variously distributed within SJ3 Ca. 'Verstraetarchaeota' population MAGs, in addition to the metabolism of hydrogen through a variety of [NiFe]-hydrogenase isoforms encoded in these genomes[56],

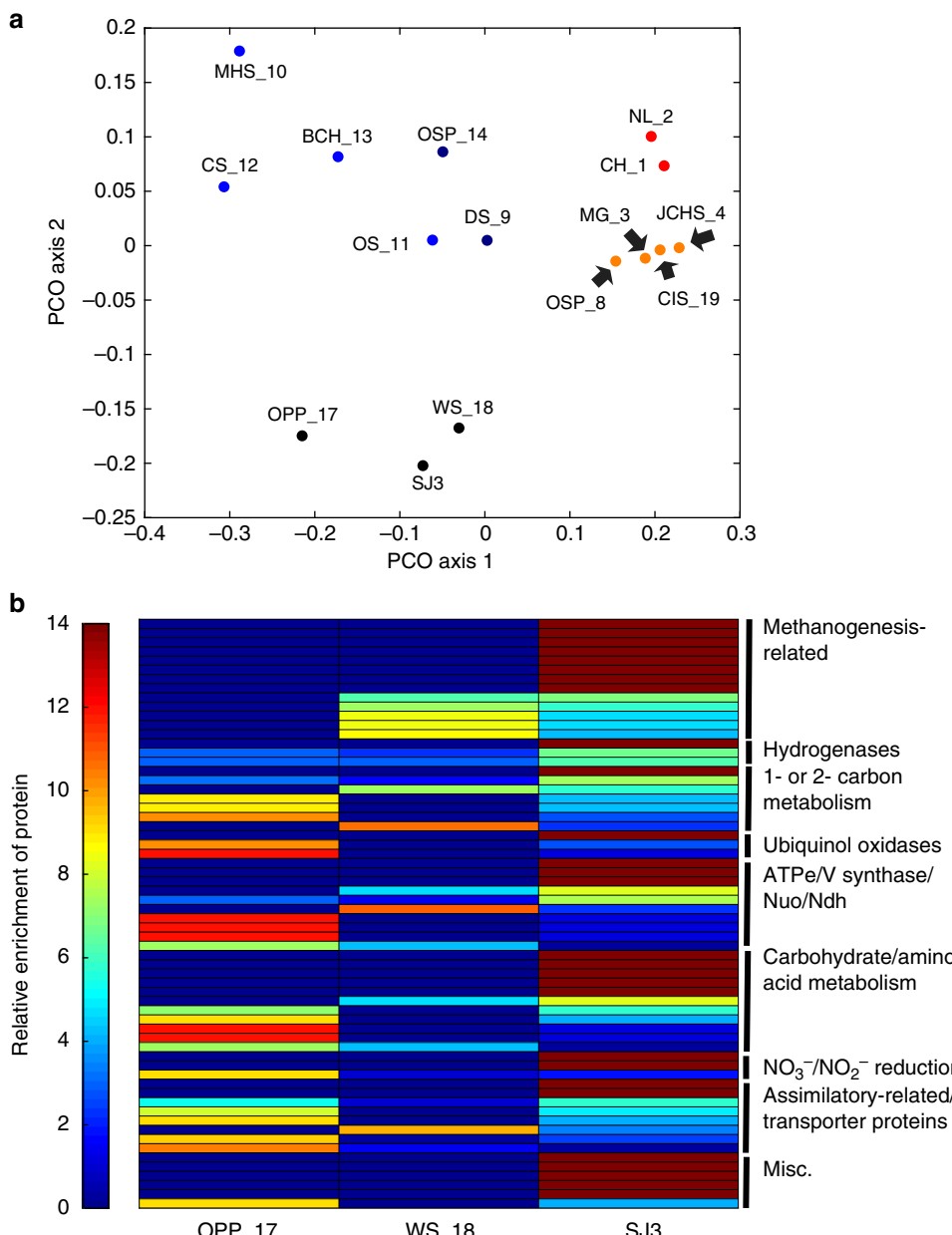

**Fig. 6** Similarity in proteins putatively involved in energy metabolism among SJ3 and 14 other published YNP metagenomes. **a** Principal coordinates analysis (PCO) shows the similarity in proteins that are putatively involved in energy metabolism pathways among metagenomes, where each point represents a metagenome. Points are colored based on taxonomic and geochemical differences: light blue, high pH Aquificales-dominated communities (pH 6.5–7.9); dark blue, low pH Aquificales-dominated communities (pH 3.1–3.5); red, low pH Sulfolobales-dominated communities (pH 2.6–4); orange, low-mid pH Crenarchaeaota-dominated communities (pH 3.4–6.4); black, mid pH Crenarchaeota/Aquificales/uncultured bacterial division dominated communities (pH 5.4–6.4). **b** Heatmap showing enrichment of specific proteins putatively involved in the energy metabolism in the SJ3 community and two other similar chemotrophic YNP thermal spring communities: OPP_17 and WS_18. The heatmap represents only the proteins that were exclusive to the SJ3, OPP_17, and WS_18 communities, and where at least one homolog was present in SJ3. Relative enrichment was calculated after normalizing annotated proteins with the total number of proteins for each metagenome to account for unequal sequencing efforts. The scale bar on the left indicates relative enrichment where a value ≥14 indicates it was exclusively found in one metagenome, relative to the other 14 considered, and a value of ≤0 indicates that the protein was not detected in the metagenome. Proteins are grouped into broader functional categories shown on the right. All protein enrichment values that are depicted here are shown in further detail in Supplementary Data 3

further indicating widely disparate physiological potential among the SJ3 *Ca.* 'Verstraetarchaeota' populations (Fig. 7e; Supplementary Data 7).

In addition to the two lineages discussed above, three phylogenetically distinct, deeply-branching Thaumarchaeota-related MAGs were recovered from SJ3 with >75% estimated completeness (and two others >50% complete). None of these

MAGs encoded proteins allowing for the oxidation of ammonium (Supplementary Data 8), a trait that has thus far typified many Thaumarchaeota cultivars and genomes[57]. Rather, these deeper-branching SJ3 thaumarchaeotes (SJ3.Bin56, SJ3.Bin70) harbored the functional capacity to consume or produce $H_2$ via the activity of [NiFe]-hydrogenase isoforms predicted to be involved in reversible $H_2$ transformation (e.g., group 3d and 4 enzymes;

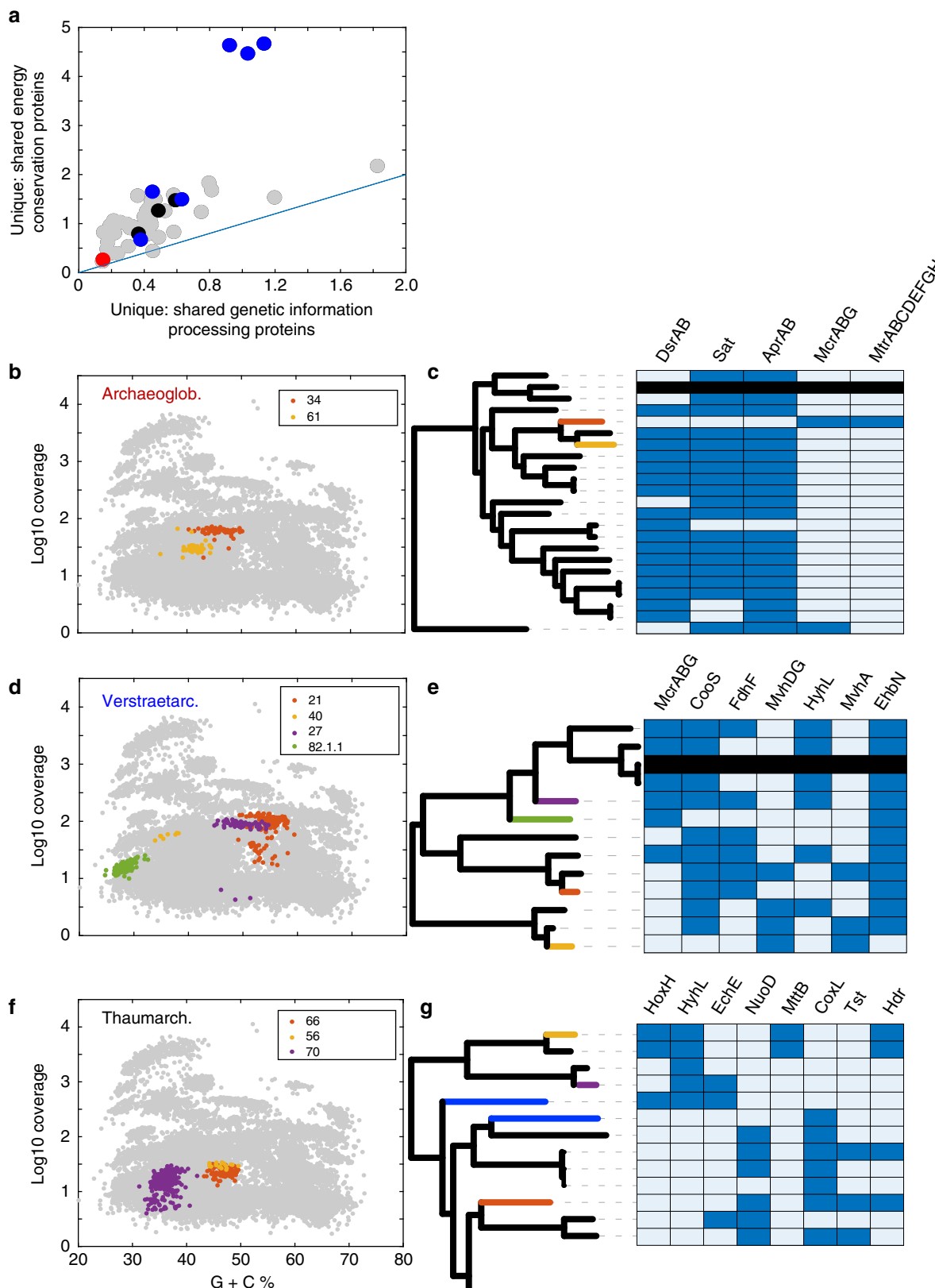

Fig. 7g; Supplementary Data 8)[56]. Conversely, genes coding for carbon monoxide dehydrogenase (CoxL) and thiosulfate sulfur-transferase (Tst) suggests the ability to use CO and $S_2O_3^-$ as electron donors, respectively, in the population represented by the SJ3.Bin66 MAG (and other later-diverging thaumarchaeotes). Genes encoding these proteins were absent in the deeper-

branching thaumarchaeote MAGs. This observation coupled with the presence of NADH dehydrogenase (Nuo) subunit homologs in later evolving thaumarchaeotes that were absent in the earlier-evolving genomes suggests an overall shift in mechanisms allowing for respiration in these lineages[58]. This may represent a transition from anaerobic metabolisms of earlier-diverging

**Fig. 7** Differentiation of energy metabolism proteins within intra-taxon genomic comparisons. **a** Ratio of unique/shared energy conservation-related proteins and the ratio of unique/shared genetic processing proteins for lineage comparisons. Each point represents a within-lineage pairwise comparison between genomes ($n = 49$). Ratios were calculated based on the number of unshared KEGG-annotated proteins relative to the number of shared KEGG-annotated proteins for the 'Metabolism' and 'Genetic Information Processing' categories. Line shows a 1:1 relationship. Differentiation in genetic information processing proteins in closely related taxa should be low, due to close phylogenetic relationships. Thus, little differentiation in metabolism-related proteins, relative to genetic processing proteins would indicate a lack of metabolic differentiation, whereas the converse would indicate greater metabolic differentiation. The Archaeoglobales, *Ca.* 'Verstraetarchaeota', and Thaumarchaeota comparisons are highlighted in red, black, and blue, respectively. Scatter plots (**b**, **d**, **f**) show the distribution of contigs based on G+C% and average sequencing coverage for genomic bins from three lineages with multiple representatives in SJ3: Archaeoglobales, *Ca.* 'Verstraetarchaeota', and Thaumarchaeota. Phylogenies and heatmaps of key differential functional genes (**c**, **e**, **g**) are shown for each of the three comparisons, respectively. Phylogenies are subsets from the full phylogenetic analysis represented in Fig. 3a, and branches are colored according to the color schemes for each lineage comparison on the left. Blue branches represent additional genomic bins from SJ3 that were estimated to be < 75% complete, and thus not included in these comparisons. Only the subset of the deepest-branches of the Thaumarchaeota are shown in **g**, while all genomes in our database are shown for the Archaeogloblaes and *Ca.* 'Verstraetarchaeota' in **c** and **e**, respectively. Protein abbreviations are as follows: DsrAB (dissimilatory sulfite reductase), Sat (sulfate adenylyltransferase), AprAB (adenosine 5′-phosphosulfate reducatase), McrABG (methyl coenzyme-M reductase), Mtr (N5-methyltetrahydromethanopterin methyltransferase), HoxH ([NiFe]-hydrogenase group 3d large subunit, LSU), HyhL ([NiFe]-hydrogenase group 3b LSU), EchE ([NiFe]-hydrogenase group 4 energy conserving hydrogenase LSU), NuoD (NADH dehydrogenase), MttB (trimethylamine--corrinoid protein co-methyltransferase), CoxL (aerobic carbon monoxide dehydrogenase LSU), Tst (thiosulfate/3-mercaptopyruvate sulfurtransferase), Hdr (heterodisulfide reductase subunits ABC), CooS (anaerobic carbon monoxide dehydrogenase catalytic subunit), FdhF (formate dehydrogenase-H), MvhADG ([NiFe]-hydrogenase group 3c methyl viologen reducing hydrogenase subunits DG), EhbN (energy converting hydrogenase B subunit N)

Thaumarchaeota to aerobic metabolisms within later-evolving Thaumarchaeota. The presence of genes encoding proteins in the earlier-evolving thaumarchaeote MAGs that are involved in methanogenesis including HdrABC (heterodisulfide reductase) and MttB (trimethylamine methyltransferase) subunits (which are involved in methylotrophic methanogenesis in the Methanomassiliicoccales, and potentially in the *Ca.* 'Verstraetarchaeota'[50,59]) further highlight these evolutionary shifts. However, the lack of other necessary protein complements for methanogenesis (e.g. Mcr, Mtr, Fwd) in these two thaumarchaeote MAGs (inclusive of SJ3.Bin56) suggests that they are unlikely to be involved in methylotrophic methanogenesis (Supplementary Data 8).

These observations, taken together, suggest that phylogenetically similar populations within SJ3 are metabolically differentiated in a manner that is likely related to selective pressure to diversify into new niches made available by the distinctive environment of SJ3 and other similar hydrothermal environments. Thus, metabolic differentiation and subsequent habitation of environmental niches created by dynamic mixing of volcanic gas-enriched hydrothermal fluids and meteoric waters may also help explain the high level of diversity that is present in SJ3 relative to other chemosynthetic hot spring communities. The data presented here suggests that much of this differentiation can be linked to diversification to take advantage of lithogenically-sourced compounds in SJ3 waters made available by variable mixing of end member geothermal fluid types that characterize hot spring environments.

## Discussion

Hydrothermal springs with pH values between 4.0–6.0 are understudied microbial ecosystems relative to other hydrothermal spring types. The pH of these springs reflects underlying fluid mixing processes wherein volcanic gas phase-influenced end member waters mix with near-surface sourced meteoric waters and/or deeply-sourced hydrothermal fluids, resulting in weakly acidic spring waters. We suggest such conditions promote and maintain extensive disequilibria in oxidation-reduction reactions that support distinctively high chemosynthetic microbial biodiversity relative to other springs that reflect minimal or no mixing of fluid types. Although little studied in continental hydrothermal systems, considerable evidence exists for the influential role of hydrothermal and seawater mixing on deep-sea vent microbial

diversity. Chemosynthetic communities within intermediate pH springs across YNP are similar in microbial taxonomic and functional compositions suggesting that such water mixing regimes generally select for and sustain unique and exceptionally diverse microbial chemosynthetic communities. Several lineages of microorganisms with inferred metabolisms not observed previously in those lineages suggest that such conditions might also even promote the emergence of new biodiversity. This metabolic differentiation functions to minimize niche overlap while maximizing the use of available nutrients (particularly gases), further promoting increased biodiversity in these settings. Taken together, these results provide new understanding of how geologic settings lead to fluid mixing dynamics that then sustains high chemosynthetic community biodiversity and may also lead to the generation of new chemosynthetic biodiversity. Moreover, these data provide important context for understanding the role of subsurface and near-surface water mixing in promoting chemosynthetic biodiversity in geothermal systems, and other similar environments, including the subsurface of Earth today and in pre-photosynthetic ecosystems of early Earth.

## Methods

**Sampling, metagenomic sequencing, and data processing**. Sediment samples for community analyses were collected from SJ3 spring on 22 July 2014. Spring temperature and pH were measured with a portable pH meter and a temperature-compensated probe (WTW 3100; WTW, Weilheim, Germany). Water conductivity was measured using a temperature-compensated probe (YSI EC300; YSI Inc. Yellow Springs, Ohio). Triplicate sediment samples (~250 mg) were collected sterilely, immediately frozen on dry ice, and transported back to the laboratory. DNA was extracted from triplicate sediment samples (~250 mg each) using the Fast DNA Spin Kit for Soil (MP Biomedicals, Irvine, CA) following the manufacturer's instructions as conducted previously[60]. Equal volumes of triplicate DNA extracts were then pooled for further analyses.

Whole community shotgun metagenomic sequencing was conducted on total genomic DNA from SJ3 sediments (~5 ng total). Paired-end sequencing (2 x 250 bp) was conducted at the Genomics Core Facility at the University of Wisconsin-Madison on the Illumina Hiseq 2500 Rapid platform. Fragmented DNA was prepared using the Illumina Nextera DNA library preparation kit (Illumina, San Diego, CA, USA) according to the manufacturer's protocols. Reads were quality trimmed and cleaved of Illumina sequencing adapters using Trimmomatic v.0.35[61] and the following parameters: LEADING:3, TRAILING:3, SLIDINGWINDOW:4:15, and MINLEN:36. MEGAHIT v.1.1.1[62] was used to assemble the quality-filtered reads into contigs using a range of k-mer values (21, 29, 39, 59, 79, 99, 119, and 141) and default parameters. MetaQUAST v.3.2[63] was then used to assess the quality of the assemblies and a final assembly was chosen using a k-mer size of 141. Raw reads were aligned and mapped to the final contigs using Bowtie2[64].

Contigs (>2.5 kbp) from the highest quality assembly (k-mer size = 141) were binned into draft metagenome-assembled-genomes (MAGs) using unsupervised binning in MetaBAT v.0.26.3[65] based on tetranucleotide frequency distribution patterns and differential sequence coverage profiles with the "verysensitive" parameter settings. Draft genome bins were then assessed for quality, contamination, and completeness using CheckM v.1.0.5[66]. Manual curation of bins was conducted using several methods to improve the quality of bins that appeared to represent multiple populations based on marker gene 'contamination' estimates. First, 'outlier' contigs that were defined as outside of 95% of the distributions for tetranucleotide word frequency distance, G+C content, or coding density of each genome bin were removed using CheckM. Second, genome bins that clearly consisted of multiple populations were re-binned using MetaBAT specifying increasingly more stringent sensitivity parameters, until 'contamination' in each bin was minimized. Contigs in each bin were also surveyed for obvious coverage or G+C% value deviations from the majority of the bin's contigs. Only medium-high quality draft genome bins are included in the analyses presented here (>50% complete, < 7% contamination).

Gene prediction and annotations were then conducted using Prodigal v.2.6.3[67] as implemented in Prokka v.1.11[68] using the default parameters, or, as implemented in CheckM using the default parameters. To compare the robustness of genome bin assignments, additional binning was performed using the unsupervised binning program CONCOCT v.0.4.1[69], and the quality of the CONCOCT bins were assessed as described above. The MetaBat- and CONCOCT-produced bins were compared against one another to assess the reliability of genome bin differentiation within and between the two methods. In some cases, the completeness of genome bins that were estimated to be abundant (>1.0% estimated relative abundance) exhibited low completeness, as is common in deep metagenomic sequencing of environmental genomic DNA. Improvement in the quality of these bins was attempted by recruiting contigs to genomes that were publicly available or were available from other in-house metagenomes and shared high marker protein identity to the low-completeness bins. Contig recruitment was conducted with the MG Wrapper tool (https://github.com/dunfieldlab/mg_wrapper) using an 80–90% identity level and 50% alignment length, followed by extraction of quality-filtered reads mapped to the contigs using the Multi-metagenome package[70]. Extracted reads were then reassembled using the Spades v.3.10.0 assembler[71], which produced higher quality assemblies than MEGAHIT for individual populations. K-mer sizes of 21,33,55,77,99,127 and the --meta and --only-assembler options were used for assembly. Contigs > 1000 bp were used for these bins, as the reference-based extraction procedure allowed for higher-quality genome assemblies. Assessment of assembly quality and population homogeneity was conducted with MetaQUAST and CheckM, and final genome bins were curated as described above using RefineM[35] to identify and filter outlier contigs from bins and to also employ k-means clustering to separate bins with multiple populations on the basis of various genomic properties (e.g., coverage profile, tetranucleotide frequencies). Relative abundances were calculated for individual populations following the methods of Hu et al.[72], wherein relativized coverage was first calculated for each bin by dividing total mapped reads by total length. This measurement was then normalized to estimated genome size by multiplying relativized coverage by estimated genome size. Finally, the relative abundance for each population was determined as the fraction represented by each population of the total summed coverage values.

**Phylogenetic analyses**. Curated genome bins were surveyed for the presence of 104 archaeal-specific or 31 bacterial-specific single copy phylogenetic marker genes with Amphora2[73]. Publicly available references from genome databases (IMG and NCBI) were downloaded based on homology searches of RpoB proteins from each bin (or ribosomal protein sequences if RpoB was not present) against each database. Only references with > 50% estimated completeness were included in the final trees. Each of the proteins were aligned individually using Clustal Omega v.1.2.0[74], and the protein alignments were concatenated into a super matrix for each domain. The concatenated alignments for Archaea ($n$ = 713; 55,322 informative amino acid positions) and Bacteria ($n$ = 497; 15,290 informative amino acid positions) were subjected to Maximum Likelihood phylogenetic analysis in RAxML v.8.2.9[75] specifying an LG protein substitution model and a Gamma shape distribution. The robustness of each clade's monophyly was assessed with 100 rapid algorithm bootstraps. Taxonomic clades were annotated based on monophyly and previously published designations. Clades without current genomic representation were identified based on monophyletic groupings and branch-lengths equal-to or greater than those used to define commonly accepted division-level designations. To assess the total branch length associated with genomes from SJ3, the branch lengths of all taxa in the final trees were calculated using the 'distRoot' function in the 'adephylo' package for R with default settings[76].

**Protein family diversity and comparison to other metagenomes**. To assess the diversity of protein diversity and functionalities in SJ3 relative to other YNP metagenomes, proteins >50 amino acids in length from SJ3 and 47 other YNP metagenomes publicly available in the IMG database as of 10/01/2017 (Supplementary Data 10) were clustered into protein family 'bins' using CD-HIT v.4.6.5[77] with the following parameter settings: -g 1 -G 0 -aS 0.8 -d 0. Communities were identified as 'chemosynthetic' based on previously published information for the

specific metagenomes or on the springs that were analyzed. The entire dataset was first clustered at the 90% homology level ($k$ = 5), followed by clustering at the 60% homology level ($k$ = 4), and a final clustering step at the 30% homology level (-ce 1e-6). Abundance-weighted protein bin counts were then calculated within each metagenome, and this dataset was used in a rarefaction analysis (i.e., random selection of increasingly larger individual proteins from each sample, and concomitant identification of total protein bin diversity) using the mothur software package[78]. Metagenomes that appeared to be derived from non-natural samples (i.e., enrichment cultures or synthetically derived communities) and/or potentially photosynthetic communities based on published or other available data were excluded from the analyses. To compare protein-coding gene diversity against environmental factors, relevant geochemical parameters (pH, temperature, $SO_4^{2-}$, $Cl^-$) were sought for each spring (Supplementary Data 10), based on publications referring either to the specific metagenome, or to other geochemical analyses for the spring. Additional information about the source of metadata for each spring is provided in Supplementary Data 10.

The functional content of SJ3 was then compared against 14 other well-characterized YNP metagenomes to identify proteins that differentiated SJ3 from other YNP springs (Supplementary Data 10). Total proteins from each of the 15 metagenomes were first annotated against the KEGG functional database[79] using the KEGG Automatic Annotation Server[80]. An abundance-weighted table of KEGG orthology (KO) assignments was then computed as described above. A subset of KOs were extracted ($n$ = 887) that encompassed proteins in the 'Energy Metabolism' subcategory of metabolism. The abundances of each KO were normalized to total metagenome KO size to account for unequal sequencing efforts between SJ3 and the much smaller, previously characterized YNP metagenomes. A Bray-Curtis distance matrix was then produced from the KO matrix using the veg. dist function in the R v.3.4.3 (https://www.r-project.org) vegan package v.2.4–6 (https://cran.r-project.org/web/packages/vegan/), which was then used in a Principal Coordinates Analysis (PCO) using the labdsv package v.1.8.0 (https://cran.r-project.org/web/packages/labdsv/) for R. The enrichment of specific KO-annotated proteins in each metagenome was assessed by first normalizing the KO abundances for each metagenome based on the total number of annotated KOs within each metagenome. The normalized data were then expressed as enrichment values where the mean for the whole dataset was subtracted from the value for each metagenome, and then normalized by the mean of the whole dataset (e.g.: [$KO_Y$ normalized abundance for Sample Y – total mean $KO_X$]/ total mean $KO_X$). The resultant value indicates the enrichment of individual KOs within each metagenome, relative to all 15 YNP metagenomes in the dataset.

**Functional comparisons among taxa**. We attempted to further identify the ecological mechanisms underlying the prevalence of numerous distinct populations of the same higher-order taxonomic lineages. Intra-lineage comparisons were first identified by assessing genomes (>75% estimated completeness only) that were present within the same monophyletic clades determined in the phylogenomic analyses. The number of shared and unshared 'Metabolism' category KOs were identified and compared against the number of shared and unshared 'Genetic Information Processing' category KOs to determine the magnitude of metabolism-related protein differences in comparison to differences of proteins involved in genetic processing. Differences in genetic processing proteins would be expected to be more conservative among closely related taxa and was thus used as a baseline. The relative numbers of protein differences for both categories were then identified across all intra-lineage comparisons ($n$ = 49 pairwise comparisons). To provide additional resolution for some pairwise comparisons, pairwise amino acid identity values[53] were calculated between genomes using the enveomics software package[81].

**Reporting summary**. Further information on experimental design is available in the Nature Research Reporting Summary linked to this article.

## Data availability

Assembled contigs have been uploaded to the Integrated Microbial Genomes (IMG) database under the genome ID 3300029625. The source data for Figs. 4 and 5 and Supplementary Fig. 3 are provided as a Source Data file.

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

## Acknowledgements

This work was supported by a Montana Space Grant Consortium grant to D.R.C. and E.S.B. and a grant from the National Science Foundation to E.S.B. and D.R.C. (EAR-1820658). M.R.L. acknowledges support from the NASA Earth and Space Science Fellowship program (NNX16AP51H). The NASA Astrobiology Institute is supported by grant number NNA15BB02A (to E.S.B.). We thank Christie Hendrix, Stacey Gunther, and Annie Carlson at YNP for research permitting.

## Author contributions

D.R.C., M.R.L., and E.S.B. were responsible for generating and analyzing the metagenomics data. D.R.C. and E.S.B. wrote the manuscript with contributions from M.R.L.

## Additional information

**Competing interests:** The authors declare no competing interests.

