## [Peer Review File · Nature Communications]

Reviewers' comments:

Reviewer #1 (Remarks to the Author):

The paper by Colman et al reports on the biodiversity of the chemosynthetic hot spring SJ3. Their metagenomic analysis revealed 1) large taxonomic diversity, 2) a similar functional potential to other chemosynthetic communities and 3) the possible existence of differentiated metabolism within closely related SJ3 populations. This led the authors to suggest that the dynamic mixing of waters generated by geological processes play a key role in the generation and maintenance of chemosynthetic biodiversity.

I am going to be blunt and start to state that from the biological point of view I had trouble following the innovation of the hypotheses. The first hypothesis states larger chemosynthetic diversity in systems where gas-rich vapor waters mix with oxidizer waters (thus creating a chemical disequilibrium and also introducing more biological relevant chemical species to the system) versus systems where this mix does not occur (systems with a narrow range of biologically relevant species). I would not consider this "a hypothesis" but common sense, specially since the characterization of SJ3 hydrothermal system is only minimally addressed here. The second hypothesis states that the organisms will use the electron donors and C supplied from the rich vapor gas input. Microorganisms tend to exploit their environment as much as they can in order to survive so if a rich-gas phase is providing them with nutrients, they will tend to use them.

I agree with the authors that these type of hydrothermal systems are still poorly understood and I recognize the high taxonomic and functional diversity from the recovered metagenomic drafts.

The most relevant finding of the paper is the possible metabolic abilities of SJ3.bin 34, in relation to the known Archaeoglobales. However, more details should be given regarding this (and other genomes) such as number of contigs as well as the mapping of the mcr complex to the contigs presented in figure 6b.

At least for the bins in figure 6, a table with the annotations and Kegg mapping should be given as supplement.

Please clarify the affiliation of bin 56 and 70, shown in figure 6f since in the Table S2 they have the "unclassified Aigararchaeota-like" affiliation so the discussion regarding their differences to JS3_bin66 does not make sense.

Minor

The colour codes in several of the panels from Sup. Figure 5 is not correct.

Please correct reference 8

Reviewer #2 (Remarks to the Author):

Major comments

The authors analyzed metagenomic sequences and numbers of metagenomic bins including previously uncultivated lineages of both Archaea and Bacteria from a hot spring sediment sample. The findings from the metagenomic bins provide novel insights into the physiology and ecological

functions of these lineages. In addition, the geochemical control of microbial diversity and function among diverse Yellow Stone National Park hot springs is also interesting. However, such geochemical impacts on geothermal ecosystems are also discussed in the parallel paper (Lindsay et al. in review), and I think the parts of biogeochemistry and geological and geochemical impacts of microbial ecosystems should be combined with the parallel paper. Due to the excessive description about geochemistry and geological settings, the information and discussion about the novel findings from the metagenomic bins are not sufficient in this manuscript. Accordingly, I recommend authors to reconstruct the manuscript to concentrate findings from the metagenomic bins in the revision process.

Specific comments

Introduction: It is better to address the issues related to the genomic and physiological features of the uncultivated lineages found in this analysis in more details.

Results and Discussion

"Geologic context of the Smokejumper geyser basin" and "Geochemical context of Smokejumper 3 spring" should be combined and shortened.

P8, L17: Taxonomic and/or physiological information about Sulfurihydrogenibium and Caldisericum would be helpful; e.g. chemolithoautotrophic, heterotrophic, Aquificales, OP5 etc.

P10, L9: "Candidatus (Ca.) Acetothermia"

P10, L19: Delete the space after "Fig.3"

P12, Table S3: Please provide relationship between the geochemical data set and metagenomic sequences with appropriate references if applicable.

L17-18: The capability of methanogenesis in Archaeoglobales has long been discussed, e.g. Stetter et al. 1987, Gorris et al. 1991, Nocek et al. 2007 etc.

Reviewer #3 (Remarks to the Author):

This is an interesting paper that reports the metagenomic analysis of a single hot spring, and compares that with other available metagenome data from the same hot spring environment, Yellowstone National Park.

Also some interesting metabolisms and findings are presented, it doesn't seem very rigorous to base the conclusions on this one sample, in comparison to a hand full of other analyses that were perhaps also not analyzed, assembled and binned in the same way. Granted the authors do argue that depth of sequencing could not account for the results seen, but it would be nice to see a statistical/ecological trend (ie from more similar sample metagenomes, rather than just one) that more mixing produces more diversity.

Small comments, but why not show data mostly with the more complete bins (except with known parasites/symbionts)? Most of your Bins are greater than 80 or 90%.

Many readers do not know what is meant by 'fluid'. I would recommend being explicit, and say geothermal.

It may be interesting to put your data in a broader context. Compare with data from deep-sea

hydrothermal vents like that of Huber and others, where mixing is also very dramatic.

This is a nice paper, and it may have a better home in a more microbiological focused journal.

Response to Reviewers Comments:

Reviewer #1 (Remarks to the Author):

The paper by Colman et al reports on the biodiversity of the chemosynthetic hot spring SJ3. Their metagenomic analysis revealed 1) large taxonomic diversity, 2) a similar functional potential to other chemosynthetic communities and 3) the possible existence of differentiated metabolism within closely related SJ3 populations. This led the authors to suggest that the dynamic mixing of waters generated by geological processes play a key role in the generation and maintenance of chemosynthetic biodiversity.

I am going to be blunt and start to state that from the biological point of view I had trouble following the innovation of the hypotheses. The first hypothesis states larger chemosynthetic diversity in systems where gas-rich vapor waters mix with oxidizer waters (thus creating a chemical disequilibrium and also introducing more biological relevant chemical species to the system) versus systems where this mix does not occur (systems with a narrow range of biologically relevant species). I would not consider this "a hypothesis" but common sense, specially since the characterization of SJ3 hydrothermal system is only minimally addressed here. The second hypothesis states that the organisms will use the electron donors and C supplied from the rich vapor gas input. Microorganisms tend to exploit their environment as much as they can in order to survive so if a rich-gas phase is providing them with nutrients, they will tend to use them.

We thank the reviewer for these comments, but respectfully differ in opinion that the hypothesis presented in the manuscript is common sense. The hypothesis that increased end member mixing would lead to increased chemical disequilibria, thereby leading to greater biological diversity has long-standing in the environmental microbiology and geobiology research community. However, we would argue that it has not been explicitly investigated in either a quantitative or semi-quantitative manner. Thus, we believe that testing this hypothesis and providing data to tease apart how this relationship would manifest in ecosystems is an important goal for environmental microbiology, and one of the primary goals of this manuscript. We have sought to decrease much of the geologically-focused background and increase discussion of the novelty of the genome bins in the manuscript in response to the editor's and other reviewer's concerns. Nevertheless, we have discussed this hypothesis, and how the data presented here are contextualized within the broader hydrothermal ecology literature in greater detail in the results sections (see lines 207-249).

Regarding the second comment from the reviewer, we emphatically agree that microorganisms are adapted (and communities assembled) to generally optimally conserve energy from the substrates that are available to them, regardless of environment. In this sense, we apologize for the confusion, but this was not the focus of our hypothesis, but rather that the mixing process would be reflected in the metabolisms of the microbial populations present in these systems. Without making this connection, it would not follow that these communities are actually assembling (and potentially adapting at the population level) to the specific geochemical characteristics associated with these mixed water environments. Indeed, because the overall metabolic profile of the community appears to reflect this mixing process, we believe that this provides further evidence that the high diversity associated with these systems is a consequence of the specific geochemical environment (resultant from mixing) that is present in these systems. Moreover, because these communities appear to be well adapted to conserve energy from their specific gas-infused environments, this provides evidence that these environments are stable enough through time (albeit not in necessarily the same location) to allow these adaptations to accrue. These sets of observations then allow it to be a reasonable

hypothesis that these geochemical conditions can lead to the generation of new biodiversity, as we have argued in the manuscript. Thus, we appreciate the reviewer's comments, but would argue that these ideas may be common sense to those in the field, but that they are far from proven with rigorous assessment in natural systems.

I agree with the authors that these type of hydrothermal systems are still poorly understood and I recognize the high taxonomic and functional diversity from the recovered metagenomic drafts.

We agree that these systems are poorly understood, despite that they are likely to be highly relevant from several ecological and evolutionary perspectives, in addition to current initiatives to understand uncultivated microbial genomic diversity. These ideas are now featured more prominently in the revised manuscript.

The most relevant finding of the paper is the possible metabolic abilities of SJ3.bin 34, in relation to the known Archaeoglobales. However, more details should be given regarding this (and other genomes) such as number of contigs as well as the mapping of the mcr complex to the contigs presented in figure 6b.

We agree with the reviewer that the evidence for putative methanogenesis in the Archaeoglobales MAG, SJ3.Bin34, is an intriguing component to the study, which is one reason we have featured this result in the manuscript. We have provided additional details about this MAG including contig mapping of the Mcr complex and other methanogenesis-related homologs (see Supplementary Fig. 5). Further, we have provided additional detail in supplementary files for all of the bins that were specifically mentioned in Figure 7 (per the reviewer comment below), including gene and contig IDs in the IMG database and KEGG mapping data (See Supplementary Files 6-8).

At least for the bins in figure 6, a table with the annotations and Kegg mapping should be given as supplement.

We have provided these data for the bins that are specifically discussed in Figure 7 in Supplementary Files 6-8. All metagenomic data is uploaded to the IMG database and will be released upon acceptance of this manuscript.

Please clarify the affiliation of bin 56 and 70, shown in figure 6f since in the Table S2 they have the "unclassified Aigararchaeota-like" affiliation so the discussion regarding their differences to JS3_bin66 does not make sense.

This was a typo in the supplemental table and it has been corrected.

Minor

The colour codes in several of the panels from Sup. Figure 5 is not correct.

We have corrected this file and removed the plots that are already shown in Figure 6 to reduce redundancy.

Please correct reference 8

This reference has been corrected.

Reviewer #2 (Remarks to the Author):

Major comments

The authors analyzed metagenomic sequences and numbers of metagenomic bins including previously uncultivated lineages of both Archaea and Bacteria from a hot spring sediment sample. The findings from the metagenomic bins provide novel insights into the physiology and ecological functions of these lineages. In addition, the geochemical control of microbial diversity and function among diverse Yellow Stone National Park hot springs is also interesting. However, such geochemical impacts on geothermal ecosystems are also discussed in the parallel paper (Lindsay et al. in review), and I think the parts of biogeochemistry and geological and geochemical impacts of microbial ecosystems should be combined with the parallel paper. Due to the excessive description about geochemistry and geological settings, the information and discussion about the novel findings from the metagenomic bins are not sufficient in this manuscript. Accordingly, I recommend authors to reconstruct the manuscript to concentrate findings from the metagenomic bins in the revision process.

We have significantly revised the scope of the paper in light of these and the other reviewer's and editor's comments. Specifically, we have reduced the focus on geochemical and geological settings in both the introduction and results/discussion, as these are indeed discussed in greater detail in our other paper under review. We have still included brief information about the geochemical setting of the spring that is the primary focus of this manuscript, as it is relevant to introduce the larger hypothesis in the paper – that mixing of geochemical end members leads to increased diversity in hydrothermal settings. To that end, we have also included additional text on the findings from the metagenomic bin analyses in accordance with this and other reviewer's comments.

We note, however, that it would not be feasible to discuss extensive details on every bin/lineage's putative physiology, as discussions of this nature are typically a full manuscript for one single lineage. Consequently, we have chosen to focus on some of the more pertinent observations from our analyses that coincide with the larger hypothesis of the research regarding hydrothermal water mixing and diversity in hydrothermal systems.

Specific comments

Introduction: It is better to address the issues related to the genomic and physiological features of the uncultivated lineages found in this analysis in more details.

Please see our response to this reviewer's comments above.

Results and Discussion

"Geologic context of the Smokejumper geyser basin" and "Geochemical context of Smokejumper 3 spring" should be combined and shortened.

These sections were combined and shortened, as suggested. We have sought to only include necessary details on the geochemical and geologic setting of SJ3 in order to provide context for the later discussion.

P8, L17: Taxonomic and/or physiological information about Sulfurihydrogenibium and Caldisericum would be helpful; e.g. chemolithoautotrophic, heterotrophic, Aquificales, OP5 etc.

This information has been included, please see lines 148-152.

P10, L9: "Candidatus (Ca.) Acetothermia"

This candidate division nomenclature has been adopted throughout the manuscript.

P10, L19: Delete the space after "Fig.3"

This has been corrected.

P12, Table S3: Please provide relationship between the geochemical data set and metagenomic sequences with appropriate references if applicable.

This information has been added in an updated version of Supplementary Table 3, along with the relevant references for the geochemical data.

L17-18: The capability of methanogenesis in Archaeoglobales has long been discussed, e.g. Stetter et al. 1987, Gorris et al. 1991, Nocek et al. 2007 etc.

Indeed, the capacity to produce small quantities of methane *in vitro* has been long-noted in the Archaeoglobales, particularly in *Archaeoglobus fulgidus* as the reviewer correctly points out is discussed in the Achenbach-Richter *et al.* 1987 paper. However, this activity must necessarily be due to an alternative mechanism to that of canonical methanogens (and the SJ3 Archaeoglobales population), since *A. fulgidus* (or other Archaeoglobales at the time of the aforementioned seminal publications) do not encode the suite of enzymes necessary for methanogenesis, and critically lack methyl-coenzyme reductase homologs that catalyze the energy conserving, methane producing step in methanogenesis. As the reviewer also correctly points out in the other papers that are cited, a number of cofactors and homologs that are critical for methanogens are also present in Archaeoglobales (e.g., the F₄₂₀ cofactor). However, these cofactors (including F₄₂₀) are present in a number of other non-methanogenic archaeal lineages (and potentially in some bacterial lineages), and appear to not be restricted to methanogens, but may rather be an ancestral archaeal trait ¹. From these observations, we would hypothesize that these are signatures of a lineage that has retained some ancestral traits in common with methanogens, but has wholesale adopted the capacity to reduce sulfate from the transfer of sulfate reducing enzymatic complements from Bacteria, as is also widely known ². We have included additional text regarding these observations within the manuscript (see lines 355-365).

Reviewer #3 (Remarks to the Author):

This is an interesting paper that reports the metagenomic analysis of a single hot spring, and compares that with other available metagenome data from the same hot spring environment, Yellowstone National Park.

Also some interesting metabolisms and findings are presented, it doesn't seem very rigorous to base the conclusions on this one sample, in comparison to a hand full of other analyses that were perhaps also

not analyzed, assembled and binned in the same way. Granted the authors do argue that depth of sequencing could not account for the results seen, but it would be nice to see a statistical/ecological trend (ie from more similar sample metagenomes, rather than just one) that more mixing produces more diversity.

We thank the reviewer for this comment and have sought to more rigorously investigate the hypothesis that increased mixing results in increased diversity. Additional analyses are included in the manuscript (see lines 207-232 and Fig. 5), that support this hypothesis. While hydrothermal fluid mixing can manifest in several geochemical outcomes, one metric that can provide insight into the extent of mixing are $\text{SO}_4^{2-}/\text{Cl}^-$ ratios of waters, which have been used in geochemical analyses to identify hydrothermal water mixing trends^{3,4}. Indeed, when the data from Fig. 4 is compared against $\text{SO}_4^{2-}/\text{Cl}^-$ ratios of numerous other YNP springs (where these data were available; Supplementary Table 3), a highly significant correlation was observed (Fig. 5). We note that mixing is certainly not the only environmental or ecological process that can lead to increased diversity within spring communities, but that it does appear to a strong contributing factor across hydrothermal springs, in general. Additional text has been devoted to discuss these new results and better contextualize the observations from this paper (lines 207-232).

Small comments, but why not show data mostly with the more complete bins (except with known parasites/symbionts)? Most of your Bins are greater than 80 or 90%.

We chose to use MAG data from those bins that were greater than 50% complete, as this has been a traditional cutoff to delineate relatively high quality metagenomic bins and is now considered the threshold for at least medium-quality MAG bins. We don't believe that this biases the analyses that were conducted considering the entire metagenomic dataset, since these data were really only used to identify taxonomic lineages that were present or otherwise conduct whole metagenome functional comparisons. The pairwise comparisons that were conducted for MAGs within the same lineage were only conducted and discussed for those MAGs that were estimated to be >75% complete in order to limit biases arising from inferred gene absence that could've likely been due to MAG incompleteness. Thus, we expect that only using the highest quality MAGs would not be particularly beneficial to the present study, nor would it dramatically alter the analyses presented here, except to underestimate the diversity of the system.

Many readers do not know what is meant by 'fluid'. I would recommend being explicit, and say geothermal.

Thank you for this suggestion. We have sought to limit the level of jargon associated with geochemical concepts and ideas through the manuscript, including the use of the term fluid.

It may be interesting to put your data in a broader context. Compare with data from deep-sea hydrothermal vents like that of Huber and others, where mixing is also very dramatic.

We agree that a broader context does represent an interesting area to connect the observations discussed here to hydrothermal communities, writ large. Indeed, the literature regarding hydrothermal/seawater mixing is more developed than that of continental settings, since high-temperature (>250°C) deep sea vent systems necessarily must feature mixing in order to be habitable. To that end, we have provided additional discussion of how these data compare to those observed at hydrothermal vents, and have specifically included some of the work that Huber and others have

published regarding mixing at hydrothermal vents and microbial diversity (see lines 234-249). To our knowledge, a quantitative or semi-quantitative comparison of mixing and levels of diversity has not been previously conducted for deep sea vents, although such a set of analyses would be highly insightful for understanding commonalities between continental and marine settings, which are likely to exist.

This is a nice paper, and it may have a better home in a more microbiological focused journal.

We thank the reviewer for their helpful comments, and believe that the manuscript is significantly improved after incorporation of their suggestions.

References

- 1 Jay, Z. J. *et al.* Marsarchaeota are an aerobic archaeal lineage abundant in geothermal iron oxide microbial mats. *Nat Microbiol* 3, 732-740, (2018).
- 2 Muller, A. L., Kjeldsen, K. U., Rattei, T., Pester, M. & Loy, A. Phylogenetic and environmental diversity of DsrAB-type dissimilatory (bi)sulfite reductases. *ISME J* 9, 1152-1165, (2015).
- 3 Nordstrom, D. K., McCleskey, R. B. & Ball, J. W. Sulfur geochemistry of hydrothermal waters in Yellowstone National Park: IV Acid-sulfate waters. *Appl Geochem* 24, 191-207, (2009).
- 4 Markussón, S. H. & Stefánsson, A. Geothermal surface alteration of basalts, Krysuvik Iceland-Alteration mineralogy, water chemistry and the effects of acid supply on the alteration process. *J Volcanol Geoth Res* 206, 46-59, (2011).

REVIEWERS' COMMENTS:

Reviewer #1 (Remarks to the Author):

The revised version of this manuscript shows improvements regarding the description of the new metagenomes and site of isolation.

The authors replied to all of the reviewers comments and I only have small comments to be addressed.

Due to the incompleteness of the metagenomes used in the comparisons, when referring to absence of genes please add a sentences related to their possible existence in the remaining of the metagenome (e.g. lines 351-354)

Also please combine the information given in supplementary tables (e.g. sheet Verstraearchaeota . bin 21 Scaff + PCG + Kegg) in one excell sheet. If grouped by bin makes it easier to read and to access the metabolic potential of each bin.

REVIEWERS' COMMENTS:

Reviewer #1 (Remarks to the Author):

The revised version of this manuscript shows improvements regarding the description of the new metagenomes and site of isolation.

The authors replied to all of the reviewers comments and I only have small comments to be addressed.

Due to the incompleteness of the metagenomes used in the comparisons, when referring to absence of genes please add a sentences related to their possible existence in the remaining of the metagenome (e.g. lines 351-354)

We have included additional text to address the possibility that protein coding gene absence within MAGs could be due to the lack of genome completeness, including in the specific instance that the reviewer points out (See lines 430-434).

Also please combine the information given in supplementary tables (e.g. sheet Verstraearchaeota . bin 21 Scaff + PCG + Kegg) in one excell sheet. If grouped by bin makes it easier to read and to access the metabolic potential of each bin.

This information has been combined as recommended by the reviewer in the revised Supplementary Data 6-8 files.